# Pulsed Radiofrequency Upregulates Serotonin Transporters and Alleviates Neuropathic Pain-Induced Depression in a Spared Nerve Injury Rat Model

**DOI:** 10.3390/biomedicines9101489

**Published:** 2021-10-16

**Authors:** Kuo-Hsing Ma, Cheng-Yi Cheng, Wei-Hung Chan, Shih-Yu Chen, Li-Ting Kao, Chun-Sung Sung, Dueng-Yuan Hueng, Chun-Chang Yeh

**Affiliations:** 1Department of Biology and Anatomy, National Defense Medical Center, Taipei 115, Taiwan; kuohsing91@yahoo.com.tw; 2Department of Nuclear Medicine, Tri-Service General Hospital, National Defense Medical Center, Taipei 115, Taiwan; steveccy60@gmail.com; 3Department of Anesthesiology, Tri-Service General Hospital, National Defense Medical Center, Taipei 115, Taiwan; whcken@gmail.com (W.-H.C.); yooyooman33@gmail.com (S.-Y.C.); 4Department of Pharmacy Practice, Tri-Service General Hospital, National Defense Medical Center, Taipei 115, Taiwan; kaoliting@gmail.com; 5Graduate Institute of Life Sciences, National Defense Medical Center, Taipei 115, Taiwan; 6Department of Anesthesiology, Taipei Veterans General Hospital, Taipei 112, Taiwan; sung6119@gmail.com; 7School of Medicine, National Yang-Ming Chiao-Tung University, Taipei 112, Taiwan; 8Department of Neurological Surgery, Tri-Service General Hospital, National Defense Medical Center, Taipei 115, Taiwan; hondy2195@yahoo.com.tw; 9Integrated Pain Management Center, Tri-Service General Hospital, National Defense Medical Center, Taipei 115, Taiwan

**Keywords:** pulsed radiofrequency, neuropathic pain, depression-like behaviors, positron emission tomography, serotonin transporters

## Abstract

Neuropathic pain (NP) is difficult to treat due to complex pathophysiological mechanisms. Pulsed radiofrequency (RRF) has been used widely with neuromodulation effect in refractory chronic pain treatment. A recent study found that PRF treatment may decrease chronic pain-related anxiety-depressant symptoms in patients, even though the mechanisms are unclear. Additionally, accumulated evidence has shown serotonin uptake is correlated with various neuropsychiatric diseases. Therefore, we investigated the effects and underlying mechanisms of PRF on depression-like behaviors, resulting from spared nerve injury (SNI)-induced NP. We examined the indexes of mechanical allodynia, cold allodynia, depression-like behavior, and blood cytokines by dynamic plantar aesthesiometry, acetone spray test, forced swimming test, and ProcartaPlex multiplex immunoassays in male Wistar rats, respectively. Serotonin transporters (SERTs) in rat brains were examined by using 4-[18F]-ADAM/PET imaging. We found that specific uptake ratios (SURs) of SERTs were significantly decreased in the brain regions of the thalamus and striatum in rats with SNI-induced NP and depression-like behaviors. Additionally, the decrease in SERT density was correlated with the development of a depression-like behavior indicated by the forced swimming test results and pronounced IL-6 cytokines. Moreover, we demonstrated that PRF application could modulate the descending serotoninergic pathway to relieve pain and depression behaviors.

## 1. Introduction

Chronic pain and clinical depression are connected in a bidirectional means and are a common comorbidity in clinic. Pain has a significant influence on anxiety and depression. Likewise, depression appears to be associated with higher sensation of pain severity and less tolerance of severe pain. Furthermore, extended exposure to pain causes mood dysregulation [1]. Neuropathic pain has been described as a type of pain that arises from primary damage to or dysfunction of either the peripheral or central nervous system. However, this kind of neuropathic pain with or without depression is difficult to relieve through pharmacological methods. Therefore, there is an important need to develop innovative drugs or techniques with better therapeutic efficacy for patients with comorbid pain and depression. To date, there are few studies about the use of nonpharmacological treatment, an alternative method, to decrease painful symptoms and depression. Comparatively speaking, pulsed radiofrequency (PRF) modulation is a valid alternative technique in which multiple medications and surgical treatments are not effective in pain control.

Recent researchers have suggested that neuroinflammation is an important etiological factor of neuropathic pain and neuropsychiatric disorders [2,3]. The regulation of pain is known to be related to preinflammatory cytokines and anti-inflammatory cytokines. Preinflammation cytokines (such as TNFα, IL-1β, and IL-6) cause pain; in contrast, anti-inflammatory cytokines (such as IL-4 and IL-10) reduce pain [4]. Neuropathic pain is inseparable from nerve cells, immune cells, and glial cells; these cells interact with each other to regulate the inflammatory response and immune mechanisms [5,6].

An increasing number of studies have shown that the absorption of serotonin is related to some neuropsychiatric diseases, such as Parkinson’s disease, depression, suicide, and drug addiction [7]. Studies have shown that serotonin transporter (SERT) and changes in serotonin are related to the development of chronic central neuropathy pain formation [8]. Using rat SERT imaging 4-[18F]-ADAM/PET for the brain, there have been many discussions on neuropsychiatric disorders, showing that SERT is involved in the regulation of these diseases [9,10,11]. Hagiwara et al. first found that the analgesic effect of PRF modulation is involved in the enhancement of the descending serotoninergic and noradrenergic system [12]. However, there have been no related articles discussing neuropathic pain using SERT imaging of the brain.

A recent human study showed that PRF treatment may reduce chronic pain-related anxiety-depressant symptoms, although the mechanisms are unclear. Therefore, the first aim of this study was to examine the analgesic and antidepressive effects of PRF on SNI-induced neuropathic pain and depression. The second aim was to use 4-[18F]-ADAM/PET in the different brain regions of rats, to explore the relationship among serotonin transporters in different brain regions after SNI-induced nerve injury and to investigate whether the analgesic and antidepressive effects of PRF are involved in the regulation of the SERT pathway. Moreover, the correlations among SERT, depression-like behaviors, and cytokines were investigated.

## 2. Materials and Methods

### 2.1. Animals

Our experimental protocol was authorized by the Animal Care and Use Committee (IACUC-18-204) of the National Defense Medical Center (Taipei, Taiwan). It was performed according to the Guide for the Care and Use of Laboratory Animals published by the National Institutes of Health (Bethesda, MD, USA). Male Wistar rats (BioLASCO, Taipei, Taiwan) weighing 200–250 g were raised separately in a setting comparable to the experimental setting with light and soft bedding on a 12 h cycle per day and night cycle. Animals had free access to water and food at any time for 7 days, for acclimatization prior to the experiment. All initiatives were made to lessen the number of animals utilized and their suffering. Rats were arbitrarily separated into four groups (*n* = 6 for each group): (i) a normal control group, (ii) a sham-operated group, (iii) an SNI group, and (iv) an SNI+PRF group. In the sham-operated group and SNI group, an RF needle was put on but no current was delivered. Animals were evaluated for cold allodynia utilizing the acetone spray test and for mechanical allodynia utilizing dynamic plantar aesthesiometer (DPA) 1 day prior to operation (baseline).

### 2.2. Experiments Schedule

This study mainly aimed to investigate the role of PRF on pain and depression-like behavioral tests and its effect on SERT changes in different brain regions after SNI. The experimental protocols are shown in Figure 1.

### 2.3. Establishment of the Neuropathic Pain Model

The SNI-induced neuropathic pain model was illustrated by Decosterd and Woolf [13]. SNI was carried out under 1.5–2% isoflurane (Halocarbon, NJ, USA) anesthesia. Then,2–4 mm ligated nerves of the common peroneal and tibial nerves were removed, while the sural nerve remained intact. The sham operation followed the same protocol but without nerve injury in the sham-operated group.

### 2.4. Pulsed Radiofrequency Therapy

All rats were randomly separated into four groups (*n* = 6 for each group). PRF was carried out through an electrocautery disk, put in a right decubitus position, and connected to the PRF generator (NeuroTherm NT1100; Morgan Innovation & Technology, Petersfield, UK). Immediately after SNI surgery, the 5-mm active tip electrode (NeuroTherm 22 GA) was placed vertically near the left sciatic nerve (0.3–0.4 cm proximal to the injury site) in the SNI+PRF treatment group. The PRF treatment at 480 kHz of stimulation mode with an output of 60V was provided at a rate of 2 Hz, 2 bursts/s, with a 20 ms duration for 6 min (3 min per session, with a 10 s intersession interval) at a temperature between 30 and 38 °C. The steps were performed in the sham and SNI groups with placement of the electrode proximal on the division of the left sciatic nerve and 0.3–0.4 cm proximal to the injury site, respectively, which were identical to those used in the PRF treatment groups but without the application of an electric current. After the procedure, the skin incision was closed with 4-O silk sutures, and animals were left alone to recover from anesthesia.

### 2.5. Pain and Depression-Like Behavioral Testing

Mechanical allodynia was analyzed by utilizing a dynamic plantar aesthesiometer (DPA; Ugo Basile, Comerio, Italy), which is an automated version of the von Frey filament that does not cause tissue damage [14,15]. According to the Kyoto method of the International Association for the Study of Pain, Basic Pain Terminology, DPA generates non-noxious responsive stimulations [16]. Rats were positioned in a specific plastic cage (25 cm/long × 10 cm/wide × 14 cm/high) with a cord mesh flooring and were seasoned to the cage for 15 min before each test session. A paw withdrawal response was generated by using an enhancing pressure using a blunt-end steel filament (0.5 mm in diameter) concentrated on the location of the sural nerve at the palmar surface area of the left ipsilateral hind paw. The force was raised from 1 to 50 g over 20 s, and was then held at 50 g for an extra 10 s; the rate of the force boost was 2.5 g/s. The threshold was taped as the force that evoked the hind paw removal reflex (the mean of 3 dimensions carried out at 5 min intervals).

Cold allodynia was established by gauging the back paw cool withdrawal feedback time to an acetone spray. Rats were put in a clear plastic cage on top of a cord mesh grid for paw accessibility. Rats were likewise adapted to the testing setting for 20 min before the measurement was started. Cold allodynia was reviewed by acetone (100 μL) that was sprayed onto the palmar surface area of the ipsilateral hind paw through the cord mesh floor using an multistepper pipette. The duration of biting, shaking, flinching, or licking habits that adhered to in a 2 min period was gauged [9,17]. Each rat was checked three times with a marginal interval of 5 min in between each examination.

Depression-like behavior was measured by using a forced swimming test (FWT) [18]. Rats were required to swim in a vertical plastic cylindrical tube (size 30 cm and also elevation 50 cm) including 30 cm of water kept at 23 °C for 15 min (pretest). Twenty-four hours later, the rats were required to swim for the 5 min test. During the 5 min, the ball game for the primary behavior of every rat in each 5 s duration was measured, and also the period of immobility rating was taped. After each swimming examination, the animal was dried with a towel, and the water was changed.

### 2.6. PET Imaging

The 4-[18F]-ADAM was manufactured through a referral study [19] and also was provided by Tri-Service General Hospital. First, 4-[18F]-ADAM (14.8–18.5 MBq; 0.5–0.6 mCi) was intravenously injected into the tail veins of rats for PET imaging. PET imaging was carried out according to references with minor adjustments [20,21]. Quickly, rats were anesthetized by passive breathing of isoflurane (5% isoflurane for induction and 2% for upkeep). One hour after tracer injection, PET imaging was carried out using a PET scanner (BIOPET 105, BIOSCAN). The energy window was set at 250–700 keV. The Fourier rebinning formula and two-dimensional filtered back-projection (ramp filter with cutoff at Nyquist regularity) were applied to rebuild the images. All pictures were evaluated using the AMIDE software application. Quantities of rate of interest (VOIs) for the striatum, midbrain, thalamus, and cerebellum were drawn manually from the reconstructed PET photos with the use of a rat brain atlas and magnetic vibration pictures [22]. The specific uptake ratio (SUR) exists as (target region-cerebellum)/cerebellum [20,21].

### 2.7. Assay of Serum Cytokines

Cytokines (IL-1β, TNF-α, and IL-6) in rat serum were measured using Rat ProcartaPlex multiplex immunoassays (Thermo Fisher Scientific, Vienna, Austria) according to the manufacturer’s protocol. Blood was collected from the tail artery one day before the SNI surgery, and on days 1, 15, and 29 according to the experimental protocol.

### 2.8. Statistical Analysis

SPSS software (version 22.0, SPSS Inc., Chicago, IL, USA) was used for the statistical tests. The results are expressed as means ± standard errors. Comparisons between means were analyzed with one-way analysis of variance. Differences between groups were analyzed post-hoc with Tukey’s honestly significant differences test (Tukey’s HSD). In addition, a repeated-measure ANOVA was performed for a repeated measures design with multiple observations that were collected from the same sample. The results were considered statistically significant if *p*< 0.05.

## 3. Results

### 3.1. PRF Treatment Alleviated Mechanical and Cold Allodynia in an SNI-Induced Neuropathic Pain Model

The mechanical and cold allodynia were tested in four groups. We first examined whether the SNI neuropathic pain model was successfully established. The SNI induced sensory symptoms, such as mechanical hypersensitivity and cold allodynia stimuli, which successfully appeared after 3 days and lasted at least 4 weeks in the SNI group (black triangle in Figure 2). Repeated-measures ANOVA indicated that there were significant differences between the therapy groups and the normal group. Compared with the SNI and SNI+PRF groups, immediate PRF therapy significantly attenuated the SNI-induced decline in the paw withdrawal threshold after SNI surgery from day 1 to day 28 (Figure 2A,B). Similarly, PRF therapy significantly reduced the withdrawal time of the acetone spray test following SNI surgery. Next, we further investigated the contralateral pain behaviors to confirm if left-side SNI impacted the other side. Therefore, the mechanical and cold allodynia of the contralateral paws were also tested (Figure 2C,D). There was no significant difference in the contralateral paw response among the four groups.

### 3.2. PET Images of 4-[18F]-ADAM in Normal Rats

Serotonin transporters (SERTs) are involved in numerous neuropsychiatric disorders. Further, 4-[18F]-ADAM/PET is a SERT-specific radioligand and has been validated for the evaluation of neuropsychiatric disorders implicated in serotonergic dysfunction [10,11,19,21]. PET images of radioactivity in normal control rat brains 60 min post injection of 4-[18F]-ADAM are shown in Figure 3. The results showed that the uptake of 4-[18F]-ADAM in the striatum and thalamus was high in the normal and sham groups. There was indiscernible 4-[18F]-ADAM uptake in the cerebellum among the four groups.

### 3.3. PET Images and the SUR of 4-[18F]-ADAM in Rats after Surgery

The distributions of 4-[18F]-ADAM in the rat brain are shown in Figure 3. On day 14 (Figure 3A), there was no significant difference among the sham, SNI, and SNI+PRF groups in any brain region. On day 28 (Figure 3B), we found that the uptake of 4-[18F]-ADAM was clearly lower in the SNI group in the striatum and thalamus regions than in the sham group. Uptake was improved after PRF therapy. The SURs of 4-[18F]-ADAM in the striatum and thalamus regions are shown in Figure 3C,D. On day 14, there was no statistical significance among the four groups. However, on day 28, there was an obvious difference between the SNI and SNI+PRF groups and no differences among the normal control, sham, and SNI+PRF groups. These data show decreased SERT expression in the striatum and thalamus regions after SNI on day 28. In contrast, immediate PRF therapy restored SERT expression in the striatum and thalamus regions on day 28. Similarly, the specific uptake ratios (SURs) of the striatum and thalamus are presented in Table 1 and Table 2.

### 3.4. Depression-Like Behavior

In the SNI-induced neuropathic pain model, depression-like behaviors have been commonly reported [23]. We established an SNI-induced neuropathic pain and depression-like behavior as previously described (Figure 2 and Figure 4). The immobility scores of rats in the FST are shown in Figure 4. On days 14 and 21, there were no statistically significant differences among the groups. However, on day 28, the immobility score of the SNI group was clearly higher than that of the sham and SNI+PRF groups. These data showed that SNI induced depression-like behaviors on day 28 and immediate PRF therapy may conquer the occurrence of depression-like behaviors after SNI.

### 3.5. Proinflammatory Cytokines in Rats

The expression of proinflammatory cytokines in the serum of rats is shown in Figure 5. On day 1, we found that the levels of IL-1 beta (Figure 5A) and TNF alpha (Figure 5B) were significantly higher in the SNI group compared to the levels in the other groups, but the level of IL-6 (Figure 5C) was not clearly different among the sham, SNI, and SNI+PRF groups. On day 29, there was an obvious difference in the level of IL-6 (Figure 5C) in the SNI group and no significant difference in the levels of IL-1 beta (Figure 5A) and TNF alpha (Figure 5B) among the three groups.

## 4. Discussion

This is the first study to explore the effects of PRF on SERT uptake using micro-PET brain images in a model of SNI-induced neuropathic pain and despair-like behaviors. In this study, we found that SNI induced long-lasting mechanical allodynia, cold allodynia, and consequent depression-like behaviors on day 28. We also demonstrated that SERT expression in the SNI neuropathic pain rat model was reduced in brain regions of the thalamus and striatum on day 28. Our results further support the use of PET imaging as a valuable diagnostic tool to reflect the impact of severe neuropathic pain on brain reorganization. Immediate PRF therapy after nerve injury exhibited a pronounced increase in SERT uptake measurements in the thalamus and striatum compared to the measurements in the SNI group on the 28th day. Additionally, PRF treatment reduced pain behaviors in the mechanical and cold allodynia tests and improved depression-like behavior in the FST compared to rats in the SNI group. Moreover, we further found that proinflammatory cytokines significantly increased serum IL-1β and TNF-α levels on the first day, and a pronounced increase in IL-6 was noted on day 29. However, after immediate PRF therapy after SNI, the above cytokines revealed no significant differences in the SNI+PRF group when compared to the sham group.

Pain has a major impact on levels of anxiety and depression. Several studies showed a high prevalence of depression and anxiety associated with chronic pain. The recent development of animal models has accelerated the studies focusing on the underlying mechanisms of the chronic pain and depression/anxiety comorbidity. Almost all of the pre-clinical studies on the levels of anxiety and depression associated with neuropathic pain were related to sciatic nerve manipulation in rodent models, using either nerve compression or section [24]. There are few studies about the use of nonpharmacological treatment alternatives that can alleviate pain and symptoms of anxiety and depression. Neuromodulation methods were shown to be valid alternative approaches when pharmacological or surgical treatments were not effective in pain control [1,12]. Neuropathic pain-induced depression is a typical comorbidity. Nevertheless, the mechanisms underlying comorbid diseases have still not been well clarified. Depression-like behaviors have been widely reported in the SNI model of neuropathic pain [23,25,26]. The results support our observations of the successful development of depression-like behavior in forced swimming tests on day 28 after SNI. In our study, we demonstrated that PRF was effective for the treatment of SNI-induced neuropathic pain and depression-like behaviors. In a recent prospective clinical study, Corallo et al. reported that PRF application in chronic pain reduces pain and provided anxiety-depressant symptom relief [1]. In a rat SNI model, Fang et al. demonstrated that PRF on the dorsal root ganglion prohibited the increased mechanical allodynia and depression-like behaviors of rats after SNI surgery [27]. However, mental distress and psychiatric symptoms, including depression, anxiety, and anger, may decrease the effectiveness of analgesic therapy, regardless of the modality of treatment [28,29]. Even with patients suffering from advanced osteoarthritis, analgesic efficacy of PRF was lower in those suffering from anxiety or depression [30].

Various studies have revealed that modifications in molecular signaling pathways, especially monoamine neurotransmitters (serotonin, dopamine, and norepinephrine), may result in a multifaceted relationship between depression and pain [29,31,32]. In a human study, Selvaraj et al. demonstrated that low SERT expression might be a state marker indicating acute depression that reduces recovery and is persistent even after recovery [33]. Another study also used decreased brain SERT expression as an indirect measure of serotonergic neuron density in living depressed patients [34]. Previous experimental evidence suggested that reduced SERT expression was found in two different stress-based animal models of depression [35]. The above studies are consistent with our findings. We found that the SERT density according to the amount of uptake of 4-[18F]-ADAM was obviously lower in the striatum and thalamus regions in the SNI group than in the sham group on day 28. Davis et al. asserted that the brain is capable of remarkable and widespread adaptive changes in response to peripheral injuries and results in the reorganization of the projections to the CNS [36]. The clinical application of PRF therapy has become an effective technique to treat different neuropathic pain conditions [1,27,37,38,39]. However, the underlying neurobiological mechanisms of PRF in neuropathic pain-induced depression remain vague. Hagiwara et al. first presumed that the analgesic impact of PRF modulation is partly mediated by an enhancement of the descending serotoninergic system [12].

After PRF therapy, SERT binding and the SUR of 4-[18F]-ADAM increased significantly compared to those of the SNI group. Therefore, from our results, we demonstrated that PRF modulates the descending serotoninergic pathway to relieve pain and reduce depression behaviors. Neuroplastic changes in the striatum, thalamus, and other different regions of the brain occur after peripheral nerve injury [40]. The role of the thalamus in chronic pain still exists in both animal and clinical experimental circumstances. Lesions or reversible local block of the lateral somatosensory thalamic nuclei have actually been revealed to reduce allodynia and hyperalgesia in SNI rats with mononeuropathy [41]. In approximately 50% of patients subjected to chronic neuropathic pain, medial thalamotomy produces 50–100% pain relief [42]. The striatum is the main source of input to the basal ganglion system. It has been postulated that the activation of striatal D2 receptors not only ameliorates neuropathic pain by inhibiting impulse discharge from pro-nociceptive neurons in the rostral ventromedial medulla, but also triggers a descending inhibitory pain pathway involving dopaminergic and serotonergic pathways to attenuate neuropathic pain [43]. Huo and colleagues found that alterations in metabolic connectivity appeared not only in the sensorimotor area directly related the contralateral to the affected limb with chronic neuropathic pain after brachial plexus injury, but also in more wide-ranging areas involving bilateral hemispheres [44]. Therefore, based on our findings, we assumed that PRF alleviates pain and reduces depression behaviors perhaps through the neuroplastic changes of certain brain regions.

The present knowledge suggests that the pathophysiology of depression may be allocated across many brain regions and networks. The exact role of the striatum in depression remains to be fully clarified [45]. Both repetitive transcranial magnetic stimulation (rTMS) and deep brain stimulation (DBS) have neuromodulatory effects, and they have been applied broadly in treatment-resistant depression (TRD) [45]. TMS mainly stimulates the dorsal prefrontal cortex, and its effectiveness is determined by downstream effects on the limbic system, such as the striatum and amygdala [46]. DBS therapy has been successfully used as a treatment for TRD and has demonstrated concurrent valuable antidepressive effects on monoamine neurotransmitters in certain target regions, such as the ventral striatum, anterior limb of the internal capsule, and inferior thalamic peduncle [47]. Thus, it is possible that neuromodulatory effects or interconnected circuits on the striatum and thalamus following rTMS or DBS could account for the mood regulation observed. After PRF therapy, the increase in SERT density and the SUR of 4-[18F]-ADAM in the striatum and thalamus paralleled the improvement of pain and depression-like behaviors, suggesting that immediate PRF therapy after SNI provides long-lasting protection against SNI-induced SERT reduction. Therefore, based on our results, we further postulate that PRF may modulate SERT expression in the striatum and thalamus and the descending serotoninergic pathway to achieve pain relief and reduce depression-like behaviors. Researchers’ understanding of the analgesic effect of the pulsed radiofrequency modality has advanced: Initially, the mechanism was a black box concept, but recent studies have reported changes in cellular ultrastructure, genetic expression, axonal firing frequency, synaptic transmission, and descending noradrenergic and serotonergic inhibitory control. From a mechanistic point of view, PRF seems primarily to modulate signaling cascades in small A and C fibers, while leaving myelinated fibers unaffected [48,49].

We also investigated the concentration of blood cytokines in rats. We found that IL-1β and TNF-α were significantly different on the first day and IL-6 were significantly different on the 29th day among the groups. We believe that the dramatic increase in IL-1β and TNF-α on the first day was probably caused by intense cascades after pronounced nerve injury-induced release of inflammatory mediators and the immune response. Additionally, the significant differences in the IL-6 concentration on day 29 were consistent withPET image changes. Accumulated evidence suggests that inflammation affects the brain and behavior; inflammatory cytokines are involved in the development of neuropsychiatric symptoms and depression [3]. Interestingly, previous studies have shown that inflammatory cytokines, including IL-6, impact SERT expression and function [3,50]. Enhanced SERT expression was associated with a significant decrease in despair-related immobility in the FST in IL-6 KO mice [32]. The above reports are also in agreement with our results, in which augmented SERT expression, reduced depression-like behaviors, and reduced IL-6 were noted in the SNI with the PRF therapy group on day 28 (day 29). Moreover, inhuman studies, baseline concentrations of circulating IL-6 cytokines may be useful for identifying subjects who will fail to respond to the current antidepressant therapy [3,51]. In addition, it has been reported that a subpopulation of patients with depression have higher levels of proinflammatory cytokines and chemokines [52].Our results are summarized in Figure 6.Therefore, based on our results, there are overlapping pathways and proceeding neurobiological processes in certain brain regions in the development of neuropathic pain and depression. Furthermore, affective dysfunction has been reported as a comorbidity of neuropathic pain, supporting the notion that pain and mood disorders may share some common pathogenetic mechanisms. The comorbidity of chronic pain and mood/anxiety disorders can be explained by the shared molecular mechanisms observed in both chronic pain and mood disorders, such as 5-HT transporter (SERT) and an imbalance of inhibitory and excitatory neurotransmission or pro-inflammatory and anti-inflammatory cytokines. However, further clinical and preclinical studies [53] are still needed to examine whether/how chronic pain modulates neural mechanisms [24].

We found that the reduction in SERT might be one of the clues indicating the development of neuropathic pain and depression after peripheral nerve injury. In the future, data obtained by following a series of changes in SERT and serum IL-6 could be used as a diagnostic tool for rats or patients who suffer from chronic refractory pain that would tend to develop depression later. Furthermore, PRF therapy may not only elicit a beneficial response in patients with refractory neuropathic pain but also in those with developing depressive-like behavior.

## 5. Conclusions

In summary, our results showed that SNI induced persistent neuropathic pain, induced despair-like behavior, and was associated with the changes in SERT imaging using micro-PET and IL-6 on. The application of 60 V PRF immediately after SNI attenuated pain behaviors and consequent depression-like behavior, and downregulated proinflammatory IL-6 cytokine levels and modulated the descending serotoninergic pathway in rats. Collectively, these results suggest that PRF therapy is a potentially encouraging approach for the treatment of neuropathic pain and depression comorbidities.

## Figures and Tables

**Figure 1 biomedicines-09-01489-f001:**
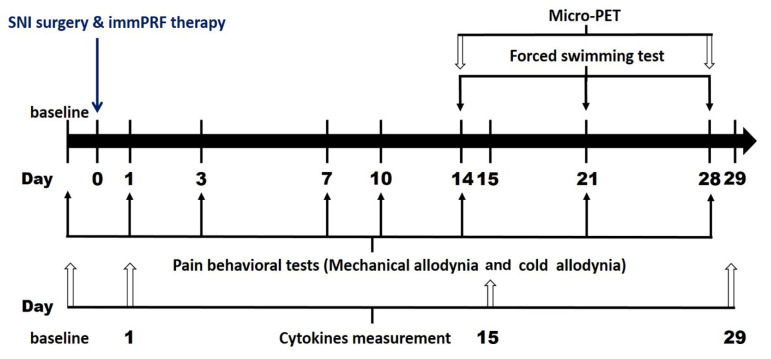
The time chart presents the treatment protocol of animal experiments in this study. During the protocol, SNI surgery and PRF therapy were performed at day 0. The behavioral testing, mechanical allodynia, and cold allodynia examination were performed on day 1, 3, 7, 10, 14, 21, and 28 of the experiment. Additionally, the rats accepted the micro-PET imaging analysis on day 14 and 28, and carried out the forced swimming test on day 14, 21, and 28. In order to avoid animal deaths caused by excessive stress, we drew animal blood for cytokines analysis 1, 15, and 29 days after entering the experiment.

**Figure 2 biomedicines-09-01489-f002:**
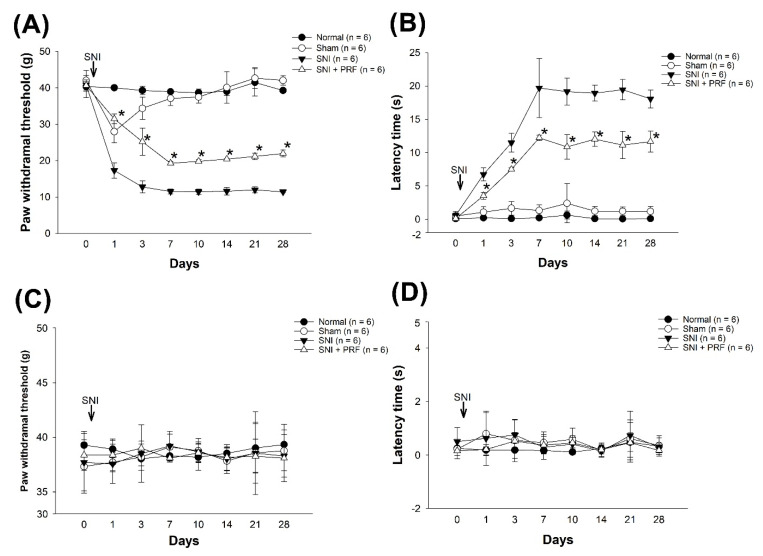
(**A**) Paw withdrawal threshold (PWT) for the mechanical allodynia test. Rats were randomly divided into 4 groups (*n* = 6): a normal control group (●), a sham group (○), an SNI group (▼), and an SNI + PRF group (△). Mechanical allodynia was examined using a dynamic plantar aesthesiometer. * *p*  < 0.05, the SNI + PRF group compared with the SNI group. (**B**) Paw withdrawal time for the cold allodynia test. Behavioral response in the 4 groups. Cold allodynia was evaluated by the acetone spray test. * *p*  < 0.05, the SNI + PRF group compared with the SNI group. (**C**) Contralateral side of PWT for the mechanical allodynia test. (**D**) Contralateral side of the paw withdrawal time for the cold allodynia test.

**Figure 3 biomedicines-09-01489-f003:**
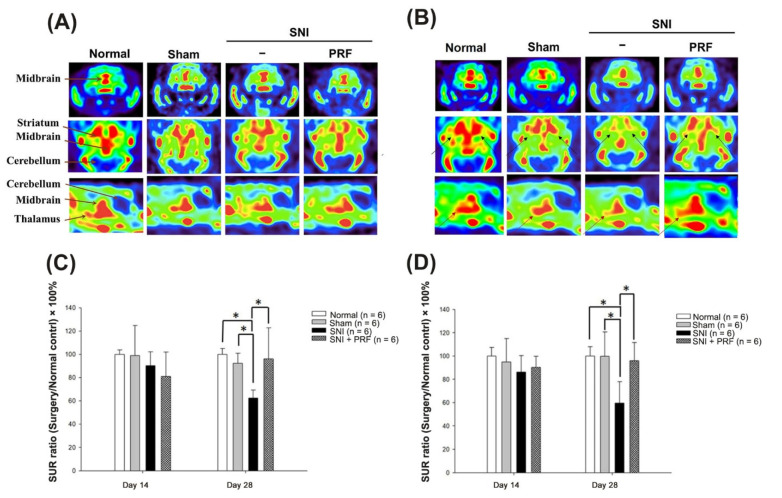
The4-[18F]-ADAM- micro-PET images and analysis of serotonin transporter levels via PET imaging. (**A**) 4-[18F]-ADAM- micro-PET images of coronal (top), transaxial (middle), and sagittal sections (bottom) of different rat brains on the 14th day after surgery. (**B**) 4-[18F]-ADAM- micro-PET images of coronal (top), transaxial (middle), and sagittal sections (bottom) of different rat brains on the 28th day after surgery. The black arrow indicates significant changes in the striatum region, and the red arrow indicates significant changes in the thalamus region. (**C**) The uptake of 4-[18F]-ADAM expressed by specific uptake ratios (SURs) was used to compare SERT levels of the striatum in the normal control, sham, SNI, and SNI + PRF groups on days 14 and 28. * *p* < 0.05, normal control, sham and SNI + PRF compared with the SNI group. (**D**) The uptake of 4-[18F]-ADAM expressed by SURs was used to compare SERT levels of the thalamus in the normal control, sham, SNI, and SNI+PRF groups on days 14 and 28. * *p* < 0.05, normal control, sham, and SNI + PRF compared with the SNI group.

**Figure 4 biomedicines-09-01489-f004:**
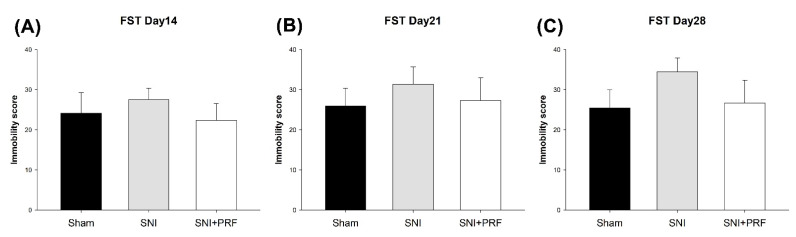
Rat immobility score on the FST test. On day 14 (**A**), day 21 (**B**), and day 28 (**C**), the immobility score on sham operation, SNI without PRF, and SNI plus PRF rats. Data are expressed as the mean ± SD. *p* < 0.05 compared to the SNI group. *n* = 6 in each group.

**Figure 5 biomedicines-09-01489-f005:**
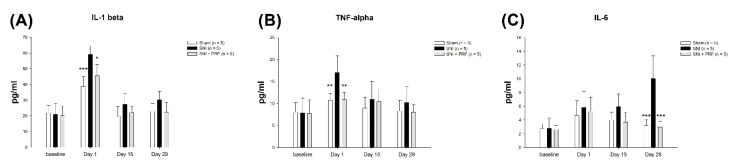
Expression of proinflammatory cytokines in serum at baseline, day 1, day 15, and day 29. The levels of (**A**) IL-1β, (**B**) TNF-α, and (**C**) IL-6 in serum of sham operation, SNI without PRF, and SNI plus PRF rats. Data are expressed as the mean ± SD. * *p* < 0.05 compared to the SNI group, ** *p* < 0.01 compared to the SNI group, *** *p* < 0.001 compared to the SNI group. n = 5 in each group.

**Figure 6 biomedicines-09-01489-f006:**
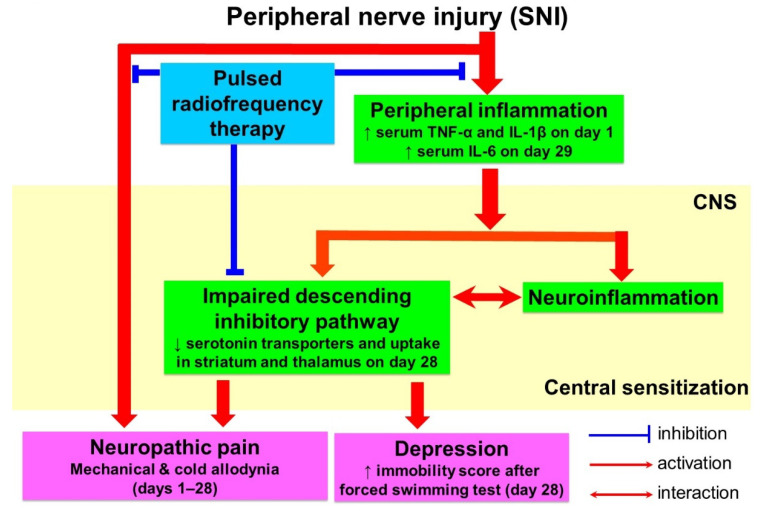
Proposed mechanism of PRF on SNI-induced neuropathic pain and depression-like behavior in rats.

**Table 1 biomedicines-09-01489-t001:** SURs in thalamus in each group.

Day after SNI	SUR (Mean ± Standard Deviation)				*p*-Value		
	Thalamus				Sham vs. SNI	SNI vs. SNI+PRF	Normal vs. Sham
	Normal	Sham	SNI	SNI + PRF			
**Day 14**	2.83 ± 0.21	2.68 ± 0.57	2.44 ± 0.40	2.55 ± 0.27	0.701	0.960	0.915
**Day 28**	2.81 ± 0.22	2.80 ± 0.59	1.67 ± 0.52	2.69 ± 0.44	0.002 *	0.005 *	1.000

SUR = specific uptake ratio; SNI = spared nerve injury; PRF = pulsed radiofrequency. *****
*P* < 0.05, sham and SN + PRF group compared with the SNI group.

**Table 2 biomedicines-09-01489-t002:** SURs in striatum in each group.

Day after SNI	SUR (Mean ± Standard Deviation)				*p*-Value		
	Striatum				Sham vs. SNI	SNI vs. SNI+PRF	Normal vs. Sham
	Normal	Sham	SNI	SNI + PRF			
**Day 14**	2.73 ± 0.10	2.70 ± 0.70	2.46 ± 0.33	2.21 ± 0.57	0.832	0.802	1.000
**Day 28**	2.78 ± 0.14	2.57 ± 0.24	1.73 ± 0.19	2.68 ± 0.74	0.010 *	0.004 *	0.802

SUR = specific uptake ratio = spared nerve injury; PRF = pulsed radiofrequency. *****
*p* < 0.05, sham and SN + PRF group compared with the SNI group.

## Data Availability

Not applicable.

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
