# Peer review of "Pulsed Radiofrequency Upregulates Serotonin Transporters and Alleviates Neuropathic Pain-Induced Depression in a Spared Nerve Injury Rat Model"

_biomedicines, 2021, doi:10.3390/biomedicines9101489_

Round 1

Reviewer 1 Report

Review of the article:

Ma K, Cheng C, Chan W, Chen S, Kao L, Sung C. Pulsed radiofrequency upregulates serotonin transporters and alleviates neuropathic pain-induced depression in spared nerve injury rat model. Biomedicines 2021:1-13.

Pain has a major impact on levels of anxiety and depression. Several studies showed a high prevalence of depression and anxiety associated with chronic pain.

The recent development of animal models accelerated the studies focusing on the underlying mechanisms of the chronic pain and depression/anxiety comorbidity.  Almost all of the pre-clinical studies on the levels of anxiety and depression, associated with neuropathic pain were related to sciatic nerve manipulation in rodent models, using either nerve compression or section.[1]

There are few studies about the use of nonpharmacological treatment alternatives that can alleviate pain and symptoms of anxiety and depression.  Neuromodulation methods were shown to be valid alternative approaches when pharmacological or surgical treatments were not effective in pain control.[2,3]

 This manuscript adds valuable data to understand how pulsed radiofrequency (PRF) treatment  could relieve pain and the associated symptoms of depression and anxiety.

However, there are some issues which merit further attention

First. Can the authors comment upon the choice of sample size of 24 rats. Furthermore, the authors should calculate the power to detect the various outcomes of interest given the sample size.

Second. The authors wrote “Therefore, based on our results, we further postulate that PRF may modulate SERT expression in the striatum and thalamus and the descending serotoninergic pathway to achieve pain relief and reduce depression-like behaviors” on page 9, sentence 351

However, as far as its analgesic effect is concerned the mechanism of action of PRF has not been completely elucidated.

Researchers’ understanding of the analgesic effect of the pulsed radiofrequency modality has advanced: Initially, the mechanism was a black box concept, but recent studies have reported changes in cellular ultrastructure, genetic expression, axonal firing frequency, synaptic transmission, and descending noradrenergic and serotonergic inhibitory control. From a mechanistic point of view, PRF seems primarily to modulate signaling cascades in small A and C fibers, while leaving myelinated fibers unaffected.[4,5]

Third. Mental distress and psychiatric symptoms, including depression, anxiety, and anger, may decrease the effectiveness of analgesic therapy, regardless of the modality of treatment.[6,7] Even with patients suffering from advanced osteoarthritis, analgesic efficacy of PRF was lower in those suffering from anxiety or depression.[8]

Fourth. Important questions which needs further to be analyzed are whether

  1. the chronic pain and mood/anxiety disorders share similar neural mechanisms or that
  2. chronic pain modulates neural mechanisms which increase the vulnerability for mood/ anxiety disorders.

The comorbidity chronic pain and mood/anxiety disorders can be explained by shared molecular mechanisms observed in both chronic pain and mood disorders such as 5-HT transporter (SERT) and imbalance of inhibitory and excitatory neurotransmission or pro-inflammatory and anti-inflammatory cytokines. However, further clinical and preclinical studies [9] are still needed to examine whether/how chronic pain modulates neural mechanisms.  [1]

REFERENCES.

[1] Humo M, Lu H, Yalcin I. The molecular neurobiology of chronic pain–induced depression. Cell Tissue Res. 2019;377(1):21-43. doi:10.1007/s00441-019-03003-z

[2] Hagiwara S, Iwasaka H, Takeshima N, Noguchi T. Mechanisms of analgesic action of pulsed radiofrequency on adjuvant-induced pain in the rat: Roles of descending adrenergic and serotonergic systems. Eur J Pain. 2009;13(3):249-252. doi:10.1016/j.ejpain.2008.04.013

[3] Corallo F, De Salvo S, Floridia D, et al. Assessment of spinal cord stimulation and radiofrequency: Chronic pain and psychological impact. Med (United States). 2020;99(3):1-4. doi:10.1097/MD.0000000000018633

[4] Van Zundert J, de Louw AJA, Joosten EAJ, et al. Pulsed and continuous radiofrequency current adjacent to the cervical dorsal root ganglion of the rat induces late cellular activity in the dorsal horn. Anesthesiology 2005;102(1):125–31.

[5] Chua NHL, Vissers KC, Sluijter ME. Pulsed radiofrequency treatment in interventional pain management: Mechanisms and potential indications - A review. Acta Neurochir (Wien). 2011;153(4):763-771. doi:10.1007/s00701-010-0881-5

[6] Rathbun AM, Stuart EA, Shardell M, et al. Dynamic effects of depressive symptoms on osteoarthritis knee pain. Arthritis Care Res 2018;70(1):80–8.

[7] Doan L, Manders T, Wang J. Neuroplasticity underlying thecomorbidity of pain and depression. Neural Plast 2015;2015:1–16.

[8] Santana Pineda MM, Vanlinthout LE, Santana Ramirez S, Zundert J van., Novalbos Ruiz JP. A randomized controlled study to compare efficacy of continuous versus pulsed radiofrequency treatment of genicular nerves to alliviate pain and improve funcional imapirment in patients with advanced osteoarthritis of the knee. Anesthesiol 2017 Reun Anu la Soc Am Anestesiólogos. 2017:A1119.

[9] Gonçalves L, Silva R, Pinto-Ribeiro F, et al. Neuropathic pain is associated with depressive behaviour and induces neuroplasticity in the amygdala of the rat. Exp Neurol. 2008;213(1):48-56. doi:10.1016/j.expneurol.2008.04.043

Review of the article:

Ma K, Cheng C, Chan W, Chen S, Kao L, Sung C. Pulsed radiofrequency upregulates serotonin transporters and alleviates neuropathic pain-induced depression in spared nerve injury rat model. Biomedicines 2021:1-13.

Pain has a major impact on levels of anxiety and depression. Several studies showed a high prevalence of depression and anxiety associated with chronic pain.

The recent development of animal models accelerated the studies focusing on the underlying mechanisms of the chronic pain and depression/anxiety comorbidity.  Almost all of the pre-clinical studies on the levels of anxiety and depression, associated with neuropathic pain were related to sciatic nerve manipulation in rodent models, using either nerve compression or section.[1]

There are few studies about the use of nonpharmacological treatment alternatives that can alleviate pain and symptoms of anxiety and depression.  Neuromodulation methods were shown to be valid alternative approaches when pharmacological or surgical treatments were not effective in pain control.[2,3]

 This manuscript adds valuable data to understand how pulsed radiofrequency (PRF) treatment  could relieve pain and the associated symptoms of depression and anxiety.

However, there are some issues which merit further attention

First. Can the authors comment upon the choice of sample size of 24 rats. Furthermore, the authors should calculate the power to detect the various outcomes of interest given the sample size.

Second. The authors wrote “Therefore, based on our results, we further postulate that PRF may modulate SERT expression in the striatum and thalamus and the descending serotoninergic pathway to achieve pain relief and reduce depression-like behaviors” on page 9, sentence 351

However, as far as its analgesic effect is concerned the mechanism of action of PRF has not been completely elucidated.

Researchers’ understanding of the analgesic effect of the pulsed radiofrequency modality has advanced: Initially, the mechanism was a black box concept, but recent studies have reported changes in cellular ultrastructure, genetic expression, axonal firing frequency, synaptic transmission, and descending noradrenergic and serotonergic inhibitory control. From a mechanistic point of view, PRF seems primarily to modulate signaling cascades in small A and C fibers, while leaving myelinated fibers unaffected.[4,5]

Third. Mental distress and psychiatric symptoms, including depression, anxiety, and anger, may decrease the effectiveness of analgesic therapy, regardless of the modality of treatment.[6,7] Even with patients suffering from advanced osteoarthritis, analgesic efficacy of PRF was lower in those suffering from anxiety or depression.[8]

Fourth. Important questions which needs further to be analyzed are whether

  1. the chronic pain and mood/anxiety disorders share similar neural mechanisms or that
  2. chronic pain modulates neural mechanisms which increase the vulnerability for mood/ anxiety disorders.

The comorbidity chronic pain and mood/anxiety disorders can be explained by shared molecular mechanisms observed in both chronic pain and mood disorders such as 5-HT transporter (SERT) and imbalance of inhibitory and excitatory neurotransmission or pro-inflammatory and anti-inflammatory cytokines. However, further clinical and preclinical studies [9] are still needed to examine whether/how chronic pain modulates neural mechanisms.  [1]

REFERENCES.

[1] Humo M, Lu H, Yalcin I. The molecular neurobiology of chronic pain–induced depression. Cell Tissue Res. 2019;377(1):21-43. doi:10.1007/s00441-019-03003-z

[2] Hagiwara S, Iwasaka H, Takeshima N, Noguchi T. Mechanisms of analgesic action of pulsed radiofrequency on adjuvant-induced pain in the rat: Roles of descending adrenergic and serotonergic systems. Eur J Pain. 2009;13(3):249-252. doi:10.1016/j.ejpain.2008.04.013

[3] Corallo F, De Salvo S, Floridia D, et al. Assessment of spinal cord stimulation and radiofrequency: Chronic pain and psychological impact. Med (United States). 2020;99(3):1-4. doi:10.1097/MD.0000000000018633

[4] Van Zundert J, de Louw AJA, Joosten EAJ, et al. Pulsed and continuous radiofrequency current adjacent to the cervical dorsal root ganglion of the rat induces late cellular activity in the dorsal horn. Anesthesiology 2005;102(1):125–31.

[5] Chua NHL, Vissers KC, Sluijter ME. Pulsed radiofrequency treatment in interventional pain management: Mechanisms and potential indications - A review. Acta Neurochir (Wien). 2011;153(4):763-771. doi:10.1007/s00701-010-0881-5

[6] Rathbun AM, Stuart EA, Shardell M, et al. Dynamic effects of depressive symptoms on osteoarthritis knee pain. Arthritis Care Res 2018;70(1):80–8.

[7] Doan L, Manders T, Wang J. Neuroplasticity underlying thecomorbidity of pain and depression. Neural Plast 2015;2015:1–16.

[8] Santana Pineda MM, Vanlinthout LE, Santana Ramirez S, Zundert J van., Novalbos Ruiz JP. A randomized controlled study to compare efficacy of continuous versus pulsed radiofrequency treatment of genicular nerves to alliviate pain and improve funcional imapirment in patients with advanced osteoarthritis of the knee. Anesthesiol 2017 Reun Anu la Soc Am Anestesiólogos. 2017:A1119.

[9] Gonçalves L, Silva R, Pinto-Ribeiro F, et al. Neuropathic pain is associated with depressive behaviour and induces neuroplasticity in the amygdala of the rat. Exp Neurol. 2008;213(1):48-56. doi:10.1016/j.expneurol.2008.04.043

Author Response

Pulsed radiofrequency upregulates serotonin transporters and alleviates neuropathic pain-induced depression in spared nerve injury rat model

RESPONSE TO REVIEWERS

REVIRWER 1

This manuscript adds valuable data to understand how pulsed radiofrequency (PRF) treatment could relieve pain and the associated symptoms of depression and anxiety.

Response: We sincerely appreciated the positive and supportive comments from reviewer to improve the quality of our revised manuscript. Our manuscript has made the necessary revisions, which were clearly highlighted in yellow.

COMMENT 1:

First. Can the authors comment upon the choice of sample size of 24 rats. Furthermore, the authors should calculate the power to detect the various outcomes of interest given the sample size.

Response: Thank you for your valuable comments. As suggested, we have used G*power to calculate the statistical power. In this study, the power was 0.91 and a power value > 0.8 is usually recognized to have an adequate sample size.

COMMENT 2:

The authors wrote “Therefore, based on our results, we further postulate that PRF may modulate SERT expression in the striatum and thalamus and the descending serotoninergic pathway to achieve pain relief and reduce depression-like behaviors” on page 9, sentence 351. However, as far as its analgesic effect is concerned the mechanism of action of PRF has not been completely elucidated.

Pain has a major impact on levels of anxiety and depression. Several studies showed a high prevalence of depression and anxiety associated with chronic pain. The recent development of animal models accelerated the studies focusing on the underlying mechanisms of the chronic pain and depression/anxiety comorbidity. Almost all of the pre-clinical studies on the levels of anxiety and depression, associated with neuropathic pain were related to sciatic nerve manipulation in rodent models, using either nerve compression or section [1]. There are few studies about the use of nonpharmacological treatment alternatives that can alleviate pain and symptoms of anxiety and depression. Neuromodulation methods were shown to be valid alternative approaches when pharmacological or surgical treatments were not effective in pain control [2,3].

Researchers’ understanding of the analgesic effect of the pulsed radiofrequency modality has advanced: Initially, the mechanism was a black box concept, but recent studies have reported changes in cellular ultrastructure, genetic expression, axonal firing frequency, synaptic transmission, and descending noradrenergic and serotonergic inhibitory control. From a mechanistic point of view, PRF seems primarily to modulate signaling cascades in small A and C fibers, while leaving myelinated fibers unaffected. [4,5]

Response: We sincerely appreciated the comments from Reviewer to improve the quality of our revised manuscript. We agreed with these important viewpoints. We have added these comments into the Discussion section of our revised text. Please see line 23-26 on Page 12, line1-6 on Page 13 and line 10-15 on Page 15.

COMMENT 3:

Mental distress and psychiatric symptoms, including depression, anxiety, and anger, may decrease the effectiveness of analgesic therapy, regardless of the modality of treatment.[6,7] Even with patients suffering from advanced osteoarthritis, analgesic efficacy of PRF was lower in those suffering from anxiety or depression.[8]

Response:

We are extremely grateful for your valuable comments. We agreed with these important viewpoints. We have added these comments into the Discussion section of our revised text. Please see line 15-19 on Page 13.

COMMENT 4:

Important questions which needs further to be analyzed are whether

  1. The chronic pain and mood/anxiety disorders share similar neural mechanisms or that
  2. Chronic pain modulates neural mechanisms which increase the vulnerability for mood/ anxiety disorders.

The comorbidity chronic pain and mood/anxiety disorders can be explained by shared molecular mechanisms observed in both chronic pain and mood disorders such as 5-HT transporter (SERT) and imbalance of inhibitory and excitatory neurotransmission or pro-inflammatory and anti-inflammatory cytokines. However, further clinical and preclinical studies [9] are still needed to examine whether/how chronic pain modulates neural mechanisms. [1]

Response Q 1+2: We sincerely appreciated the comments from Reviewer to improve the quality of our revised manuscript. We agreed with these important viewpoints. We have added these comments into the Discussion section of our revised text. Please see line 3-10 on Page 16.

We have sent our manuscript for English edition (Inovice #SFLNP7RB8) as follows.

We sincerely appreciated the valuable comments from the Reviewer 1 to improve the quality of our revised manuscript.

Reviewer 2 Report

This paper presents an interesting study of the analgesic and antidepressive effects of PRF on SNI-induced pain. Additional aims were to investigate the relationship between serotonin transporters and the brain of the SNI rats. The paper is well written and the study design is appropriate. However, the methodology requires some clarification. The selection of the behavioral test needs more clarification. The authors should provide the reason for omitting hyperalgesia and heat tests. Also, the authors should provide information on PRF procedure, the properties of the disk, the positioning of the electrode, procedures to keep it in place, control of temperature (the exact temperature should be provided), and rationale for selecting PRF settings. The authors should also comment on the unusually long effect of PRF therapy that is still present after more than three weeks.
The panels B, C, and D in figure 2 should be redrawn with the X-axis passing through 0. Also, the Y-axis should be presented without interruptions.
In conclusion, revision is required before this manuscript would be acceptable for publication.

Author Response

Pulsed radiofrequency upregulate sserotonin transporters and alleviates neuropathic pain-induced depression in spared nerve injury rat model

RESPONSE TO REVIEWERS

Reviewers’ comments from REVIEWER 2

This paper presents an interesting study of the analgesic and antidepressive effects of PRF on SNI-induced pain. Additional aims were to investigate the relationship between serotonin transporters and the brain of the SNI rats. The paper is well written and the study design is appropriate.

Response: We sincerely appreciated the valuable comments from the Reviewer 2 to improve the quality of our revised manuscript. We have fully responded this comment in Materials and methods (Q1-3), and Figure Panels (Q5) sections of revised manuscript. Our manuscript has made the necessary revisions, which were clearly highlighted in green.

  1. However, the methodology requires some clarification. The selection of the behavioral test needs more clarification.

Response: We appreciated the comment from the reviewer to improve the quality of our revised manuscript. We have followed the reviewer’s comment to clarify the selection of the behavioral test.

  1. The authors should provide the reason for omitting hyperalgesia and heat tests.

Response: We appreciate the valuable suggestion from the reviewer. In the present study, we examined the indexes of mechanical allodynia and thermal hyperalgesia (cold) by dynamic plantar aesthesiometry and acetone spray test, which is as the same as our previous studies applying PRF therapy in spared nerve injury (SNI) model [1-2]. The SNI-induced neuropathic pain model was first illustrated by Decosterd and Woolf [3]. They found that behavioral changes of mechanical (von Frey and pinprick) sensitivity and thermal (hot-radiant heat stimulus and cold-acetone spray) responsiveness is increased in the ipsilateral sural territory and are similar to clinical neuropathic pain condition [3]. We sincerely thank the reviewer’s suggestion, and we will conduct further study to use the heat hyperalgesia test to investigate pain behaviors.

Reference:

[1] Yeh CC, Sun HL, Huang CJ, Wong CS, Cherng CH, Huh BK, Wang JS, Chien CC. Long-Term Anti-Allodynic Effect of Immediate Pulsed Radiofrequency Modulation through Down-Regulation of Insulin-Like Growth Factor 2 in a Neuropathic Pain Model. Int J Mol Sci. 2015 Nov 13;16(11):27156-70.

[2] Yeh CC, Wu ZF, Chen JC, Wong CS, Huang CJ, Wang JS, Chien CC. Association between extracellular signal-regulated kinase expression and the anti-allodynic effect in rats with spared nerve injury by applying immediate pulsed radiofrequency. BMC Anesthesiol. 2015 Jun 16;15:92.

[3] Decosterd, I.; Woolf, C.J. Spared nerve injury: an animal model of persistent peripheral neuropathic pain. Pain 2000, 87, 149-158.

  1. Also, the authors should provide information on PRF procedure, the properties of the disk, the positioning of the electrode, procedures to keep it in place, control of temperature (the exact temperature should be provided), and rationale for selecting PRF settings.

Response: We appreciate the valuable suggestion from the reviewer. We have follow up the reviewer’s comment to clarify the information on PRF treatment procedure in Materials and methods section. Please see Line 17-27 on Page 5 as followings:

According to our previous studies [2], PRF was carried out through an electrocautery disk, put in a right decubitus position and connected to the PRF generator (NeuroTherm, NT1100, UK). Immediately after SNI surgery, the 5 mm active tip electrode (NeuroTherm 22 GA) was put vertically and nearby the left sciatic nerve (0.3–0.4 cm proximal to the injury site) in SNI+PRF treatment group. The PRF treatment at 480 kHz of stimulation mode with an output of 60V was provided at a rate of 2 Hz, 2 bursts/s, with a 20 ms duration for 6 min (3 min per session, with a 10 s intersession interval) at a temperature between 30~38℃. The steps performed in the sham and SNI groups with placement of the electrode proximal to the division of left sciatic nerve and 0.3-0.4 cm proximal to injury site, respectively, which were identical to those used in the PRF treatment groups, but without application of an electric current.

(From Reference [2]). Application of pulsed radiofrequency (PRF) in a model of spared nerve injury (SNI). a. Diagram. b. Actual application of PRF immediately after SNI surgery, the tip of the RF needle is a single 5-mm active electrode, which is placed adjacent to targeted nerves.

  1. The authors should also comment on the unusually long effect of PRF therapy that is still present after more than three weeks.

Response: In our previous study, we have demonstrated that immPRF-60V (immediate PRF-60V therapy after SNI) inhibited mechanical allodynia for much longer than postPRF-60V (PRF-60V on the 14th day after SNI). The modulation of immPRF treatment should imply a unique molecular mechanism which mainly through the microglial pathway. The immPRF treatment produced a long-term anti-allodynic effect by down-regulation of IGF2 and ERK activity through microglial and neuronal cells [1].

Reference

[1] Yeh CC, Sun HL, Huang CJ, Wong CS, Cherng CH, Huh BK, Wang JS, Chien CC. Long-Term Anti-Allodynic Effect of Immediate Pulsed Radiofrequency Modulation through Down-Regulation of Insulin-Like Growth Factor 2 in a Neuropathic Pain Model. Int J Mol Sci. 2015 Nov 13;16(11):27156-70.

  1. The panels B, C, and D in figure 2 should be redrawn with the X-axis passing through 0. Also, the Y-axis should be presented without interruptions.

In conclusion, revision is required before this manuscript would be acceptable for publication.

Response: We thank the reviewer’s comment. We have followed up the reviewer’s comment to revise the figure 2. Please see Page 8.
